# The Crosstalk between Cardiac Lipotoxicity and Mitochondrial Oxidative Stress in the Cardiac Alterations in Diet-Induced Obesity in Rats

**DOI:** 10.3390/cells9020451

**Published:** 2020-02-17

**Authors:** Sara Jiménez-González, Gema Marín-Royo, Raquel Jurado-López, María Visitación Bartolomé, Ana Romero-Miranda, María Luaces, Fabián Islas, María Luisa Nieto, Ernesto Martínez-Martínez, Victoria Cachofeiro

**Affiliations:** 1Department of Physiology, Instituto de Investigación Sanitaria Gregorio Marañón (IiSGM), School of Medicine, Universidad Complutense, 28040 Madrid, Spain; saraji02@ucm.es (S.J.-G.); gemarin@ucm.es (G.M.-R.); rajurado@ucm.es (R.J.-L.); anarom12@ucm.es (A.R.-M.); 2Department of Immunology, Ophthalmology and Oto-Rhino-Laringology, Faculty of Psychology, Universidad Complutense, 28223 Madrid, Spain; mvbartol@ucm.es; 3Cardiology Department, Cardiovascular Institute, Hospital Clínico San Carlos, 28040 Madrid, Spain; maria.luaces@salud.madrid.org (M.L.); fabislas@ucm.es (F.I.); 4Instituto de Biología y Genética Molecular, CSIC-Universidad de Valladolid, 47003 Valladolid, Spain; ml.nieto@csic.es; 5Ciber de Enfermedades Cardiovasculares (CIBERCV), Instituto de Salud Carlos III, 28029 Madrid, Spain

**Keywords:** cardiac fibrosis, cardiac hypertrophy, cardiac lipotoxicity, mitochondrial function, obesity, oxidative stress

## Abstract

The impact of the mitochondria-targeted antioxidant MitoQ was evaluated in the cardiac alterations associated with obesity. Male Wistar rats were fed either a high fat diet (HFD, 35% fat) or a standard diet (CT, 3.5% fat) for 7 weeks and treated with MitoQ (200 µM). The effect of MitoQ (5 nM) in rat cardiac myoblasts treated for 24 h with palmitic acid (PA, 200 µM) was evaluated. MitoQ reduced cardiac oxidative stress and prevented the development of cardiac fibrosis, hypertrophy, myocardial ^18^-FDG uptake reduction, and mitochondrial lipid remodeling in HFD rats. It also ameliorated cardiac mitochondrial protein level changes observed in HFD: reductions in fumarate hydratase, complex I and II, as well as increases in mitofusin 1 (MFN1), peroxisome proliferator-activated receptor gamma coactivator 1-alpha, and cyclophilin F (cycloF). In vitro, MitoQ prevented oxidative stress and ameliorated alterations in mitochondrial proteins observed in palmitic acid (PA)-stimulated cardiac myoblasts: increases in carnitine palmitoyltransferase 1A, cycloF, and cytochrome C. PA induced phosphorylation of extracellular signal-regulated kinases and nuclear factor-κB p65. Therefore, the data show the beneficial effects of MitoQ in the cardiac damage induced by obesity and suggests a crosstalk between lipotoxicity and mitochondrial oxidative stress in this damage

## 1. Introduction

Obesity, which is characterized by excessive accumulation of adipose tissue and lipids, forming ectopic fat deposits in different tissues including the heart [1,2], is associated with a higher risk of cardiovascular morbidity and mortality [3,4]. This cardiac lipotoxicity can trigger cellular alterations, since the exposure of the heart to elevated plasma levels of free fatty acids (FFAs) may cause accumulation of toxic lipid intermediates within cardiomyocytes [1,5]. In addition, lipids are essential regulators of cardiac function through their role in signaling, membrane structure, and transport, and as substrate for β-oxidation in the mitochondria [1,6,7]. Although fatty acid β-oxidation is the main energy source, the heart also obtains energy from glucose and/or lactate depending upon substrate availability. However, during obesity, the heart is characterized by a decrease in glucose utilization due to an excess of fatty acids (FA). This shift could reduce cardiac efficiency since FA oxidation consumes more oxygen per ATP molecule produced than glucose [8], and affects mitochondrial function. The altered insulin signaling of the obese heart is considered an adaptation to a calorie overabundance [9].

The heart demands an elevated quantity of energy being supported by a highly active mitochondrial system. Mitochondria comprise almost a third of cardiomyocyte volume, are the source of ATP, and play a central role in the calcium homeostasis necessary for maintaining cardiomyocyte viability and contractility. Mitochondria are also the main source of reactive oxygen species (ROS), which can be exacerbated in different pathological conditions, including obesity [10,11]. Dysregulation of mitochondrial function in the context of obesity can participate not only in the cardiac consequences of obesity, but as well as in its metabolic ones, since the administration of a mitochondrial antioxidant was able to improve cardiac damage and improve insulin resistance in adipose tissue associated with obesity [10,12,13,14]. Although the potential role of cardiolipotoxicity and mitochondrial oxidative stress in the cardiac damage associated with obesity is accepted, how these two mechanisms interact in this damage remains unclear. Therefore, our aim has been to characterize the interactions between cardiolipotoxicity and mitochondrial oxidative stress in the cardiac alterations in diet-induced obese rats. To address this aim, we evaluated the impact of mitochondrial lipid profile, in cardiac fibrosis and hypertrophy, as well as the cardiac metabolic alterations in obese rats. In addition, the effect of the mitochondrial-targeted antioxidant MitoQ on these changes and their consequences on mitochondrial proteins has been evaluated. We have also analyzed the effect of palmitic acid (PA) on mitochondrial oxidative stress and mitochondrial proteins in cultured cardiac myoblasts.

## 2. Material and Methods

### 2.1. Animal Model

Male Wistar rats of 150 g (Envigo, Barcelona, Spain) were fed either a high fat diet (HFD, 35% fat; Envigo Teklad no. TD.03307, Haslett, MI, USA; *n* = 16) or a standard diet (CT, 3.5% fat; Envigo Teklad no.TD.2014, Haslett, MI, USA; *n* = 16) for 7 weeks. Half of the animals of each group received the mitochondrial antioxidant MitoQ (200 µM) in the drinking water for the same period. The dose of MitoQ was based on previous data [15]. MitoQ was provided by MP Murphy from Medical Research Council Mitochondrial Biology Unit, Cambridge BioMedical Campus, Cambridge, UK. Body weight was measured once per week. Systolic blood pressure (SBP) was estimated end-of-study through use of a tail-cuff plethysmograph (Narco Bio-Systems) in unrestrained rats. The Animal Care and Use Committee of Universidad Complutense de Madrid approved all experimental procedures according to the Spanish Policy for Animal Protection RD53/2013, which meets the European Union Directive 2010/63/UE. At the end of the experiment, serum and plasma were collected.

### 2.2. Evaluation of Cardiac Structure and Function

#### 2.2.1. Echocardiographic Study

Cardiac function was evaluated by transthoracic echocardiography with an Acuson Sequoia 256 (Siemens Medical Solutions, Germany) connected to a 15-MHz linear transducer. The 2D-guided M-mode recordings were made from short-axis views to measure left ventricular (LV) chamber dimensions, interventricular septum and posterior wall thickness. LV diastolic (LVDA) and systolic (LVSA) areas were measured from the bidimentional parasternal long-axis view. The mean measurements from several consecutive beats were used for data analysis. LV chamber volumes were calculated using the cylindrical model. This model assumes the ventricle is approximated by a cylinder. Left ventricular ejection fraction (EF) was calculated in percentage according to the Teicholz Formula: (EDD^3^ × 7)/(2.4 + EDD). Fractional shortening was calculated according to the following formula: (EDD − EDS)/EDD in percentage. The diastolic function was assessed by early and late transmitral peak diastolic flow velocity (E and A waves) and ratio between E-waves and A-wave (E/A) was calculated. The images were processed with the software MASS (Medis Medical Imaging Systems, Best, Norway).

#### 2.2.2. Morphological and Histological Evaluation

Cardiac tissue samples were dehydrated, embedded in paraffin, and cut in 4 μm-thick sections. Sections were stained with picrosirius red in order to detect collagen fibers, or hematoxylin and eosin to measure cardiac myocyte area. The area of cardiac interstitial fibrosis was identified as the ratio of interstitial fibrosis or collagen deposition to the total tissue area after excluding the vessel area from the region of interest. For each sample, 10 to 15 fields were analyzed with a 40× objective under transmitted light microscopy (Leica DM 2000; Leica AG, Germany). Myocytes (60–80 per animal) with visible nucleus and intact cellular membranes were chosen for determination of cross-sectional area in hematoxylin and eosin staining sections with a 40× objective under transmitted light microscopy (Leica DM 2000). Quantitative analysis was performed using an analysis system (Leica LAS 4.3).

#### 2.2.3. Isolation of Cardiac Mitochondria

Cardiac mitochondria were isolated as reported [5]. Frozen hearts were placed and washed in cold homogenization medium containing 0.075 mol/L sucrose, 1 mmol/L EDTA, 10 mmol/L Tris-HCl, and pH 7.4. Briefly, heart tissue was homogenized (1:10 *w*/*v*) at 800br pm in a homogenizer (T 10 basic Ultra-turrax, Ika-Werke; Germany). The homogenates were centrifuged at 1300× *g* for 5 min at 4 °C to remove nuclei and debris. Supernatants were separated and centrifuged at 12,000× *g* for 10 min at 4 °C. The resulting pellets were suspended in homogenization medium and centrifuged twice at 14,400× *g* for 3 min at 4 °C to wash the mitochondrial fraction. Mitochondrial pellets were stored at −80 °C until use. Protein concentration was determined by the Bradford method.

#### 2.2.4. Lipidomic Analysis

Mitochondrial myocardial lipids were extracted and analyzed by ultrahigh performance liquid chromatography coupled to time-of-flight mass spectroscopy (UPLC-QToF-MS) using an Acquity UPLC System and a SYNAPT HDMS G2 (Waters, Manchester, UK) with electrospray ionization. Methanol:chloroform (1:2) cardiac extracts were evaporated to dryness and the pellet resuspended in 250 µL of acetone:2-propanol:ethanol (3:4:3) and used for triglyceride (TG) measurement, as previously reported [5]. Another aliquot was evaporated to dryness and the pellet resuspended in 200 µL methanol:water (9:1) and used for phospholipid (PPLs) measurement as previously reported [16]. Extracts were kept at −80 °C until analysis.

Using Lipid Maps, Metlin, CheBI, LipidBank, and KEGG databases, a particular elemental composition was searched for possible known compounds. Where possible, acyl chains were aimed at being identified by data from the high-energy function (fragmentation). To assess the specific location of each acyl chain (at the positions sn-1 or sn-2 of the glycerol backbone) is not possible using this methodology; thus, the most current structure is indicated. The chromatographic peak area from the extracted ion chromatogram of every *m*/*z* value detected, whether or not having been identified, was quantified using the QuanLynx application.

Chromatograms and mass spectra of all samples were processed with a MarkerLynx method in order to search for differential features (retention time *m*/*z*) amongst sample groups. Five injections of methanol were used as blanks to determine features prone to rejection. Only features that appeared in 66% of the samples were accepted. Sets of about 1650 features for data in negative mode and about 2109 features for data in positive mode were detected using the MarkerLynx application. They were checked manually to remove all the features that were present in the blanks. The array resulting from this process, which is comprised of samples and features as independent variables, and the feature signal intensity as dependent variable was submitted to multivariate statistical analysis using the Extended Statistics application that is available with the instrument software; this application is licensed from part of the statistical software SIMCA+ from Umetrics Ltd. (Sweden).

#### 2.2.5. In Vivo PET-CT Imaging to Study Uptake of 18F-Fluorodeoxyglucose in the Heart

Myocardial metabolic activity was evaluated by means of a small-animal dedicated dual scanner (Albira PET/CT scanner, Bruker NMI, Valencia Spain). One week before the end of the experiment, animals were starved for 18 h, followed by intraperitoneal injection with ^18^F-fluorodeoxyglucose [FDG; 12.99 ± 0.04 MBq in 0.2 mL NaCl (0.9%); Instituto Tecnológico PET, Madrid, Spain]. Twenty min later, rats underwent Positron Emission Tomography (PET) and computed tomography (CT) scanning under isoflurane anesthesia. The acquired PET and CT images were then reconstructed by, respectively, using maximum-likelihood (ML) expectation-maximization (EM) [ML-EM] and filtered-back projection algorithms. In order to account for the weight difference in rats and the [^18^F] doses of injected FDG, we calculated the standardized uptake value (SUV). The semi-quantitative SUV measurement is the most widely used in [^18^F] FDG PET studies of both small animals and humans [17]. The software used was PMOD 3.6 (PMOD Technologies Ltd., Zurich, Switzerland).

To quantify the metabolic activity, the CT image of the heart from each animal was registered to its corresponding PET image. Then a three-dimensional region of interest (ROI) was drawn on the fused PET/CT image to measure the metabolic activity of the entire left ventricle. These steps were carried out by using PMOD 3.0. SUV was obtained as an index of regional metabolic activity. The SUV parameter was calculated as a ratio of the ROI radioactivity concentration (kBq/mL) measured by the scanner, and the administered dose (kBq) was decay-corrected at the time of the injection, divided by the body weight (g).

#### 2.2.6. Measurement of Cardiac Reactive Oxygen Species (ROS) Production

For detection of either total or mitochondrial superoxide anion (O_2_^−^) levels, cardiac sections (14 µm) were incubated with dye dihydroethidium (DHE; 10^−5^ M) for 30 min or MitoSox (5 × 10^−6^ M) for 10 min, respectively at 37 °C. DHE is oxidized by superoxide and exhibits red fluorescence. MitoSOX™ is rapidly and selectively targeted to the mitochondria where it is oxidized by superoxide and exhibits red fluorescence. The oxidation products become highly fluorescent upon binding to nucleic acids. Fluorescent signals were viewed using a fluorescent laser scanning microscope (40× objective in Leica DMI 3000 microscope). Quantitative analysis of O^2−^ production was performed by using an image analyzer (Leica LAS 4.3; Leica AG, Germany). Three sections per animal were quantified and averaged for each experimental condition. The mean fluorescence densities in the target region were analyzed. Results are expressed as n-fold increase over the values of the control group in arbitrary units.

#### 2.2.7. Intracellular Lipid Droplet Evaluation

The lipid content within the cells was assessed by oil red O staining OCT frozen cardiac sections mounted on 10% glycerol in PBS-DAPI (5 ng/μL) to visualize the nucleus, and stained with Oil red O. Fluorescent signals were viewed using a fluorescent laser scanning microscope (40× objective in Leica DMI 3000 microscope). Quantitative analysis of lipid droplet content was performed by using an image analyzer (LEICA Q550 IWB Leica LAS 4.3; Leica AG, Germany). Three sections per animal were quantified and averaged for each experimental condition. The mean fluorescence densities in the target region were analyzed. Results are expressed as the ratio of lipid droplet content to the total tissue area.

### 2.3. Western Blot Analysis

Total cardiac and cellular proteins were separated by SDS-PAGE on 4–15% polyacrylamide gels and transferred to Hybond-c Extra nitrocellulose membranes (Hybond-P; Amersham Biosciences, Piscataway, NJ, USA) with the Trans-Blot Turbo Transfer System. Membranes were probed with primary antibody for acetyl CoA synthetase (ACS, Abcam; Cambridge, UK; dilution 1/500), adipose triglyceride lipase (ATL, Abcam; Cambridge, UK; dilution 1/1000), carnitine palmitoyltransferase 1A (CPT1A, Abcam; Cambridge, UK; dilution 1/1000), cyclophilin F (cycloF, Santa Cruz Biotechnology; Dallas, TX, USA; dilution 1/500), cytochrome C (cytoC, Abcam; Cambridge, UK; dilution 1/500), diacylglycerol O-acyltransferase 1 (DGTA1, Abcam; Cambridge, UK; dilution 1/1000), Dynamin-1-like protein (DRP1, Abcam; Cambridge, UK; dilution 1/500), extracellular signal-regulated kinases (Erk1/2, ThermoFisher; Rockford, IL, USA; dilution 1/1000), Erk1/2 (phosphor-p44/42 Thr202/Tyr204, Cell Signaling, Danvers, MA, USA; dilution 1/1000), fatty acid translocase (FAT; Abcam; Cambridge, UK; dilution1/1000), fumarate hydratase (FH; Santa Cruz Biotechnology; Dallas, TX, USA; dilution 1/1000), mitofusin 1 (MFN1, Abcam; Cambridge, UK; dilution 1/1000), nuclear factor-κB p65(NF-κB, Abcam; Cambridge, UK; dilution 1/1000), NF-κB (phospho S536, Abcam; Cambridge, UK; dilution 1/1000), peroxisome proliferator-activated receptor gamma coactivator 1-alpha (PGC-1α, Abcam; Cambridge, UK; dilution 1/1000), protein kinase B (Akt, Cell Signaling, Danvers, MA, USA; dilution 1/1000), Akt, (phospho Ser473, Cell Signaling, Danvers, MA, USA; dilution 1/1000), 4-hydroxynonenal (4HNE, Abcam; Cambridge, UK; dilution 1/250) and GAPDH and β-actin (Sigma-Aldrich; dilution 1/5000), as loading controls. Signals were detected using the ECL system (Amersham Pharmacia Biotech). Results are expressed as an n-fold increase over the values of the control group in densitometric arbitrary units.

The Mitoprofile Total OXPHOS Rodent WB Antibody Cocktail (Abcam¸ Cambridge, UK; dilution 1/1000) was used to quantify the relative levels of the subunits of the mitochondrial OXPHOS complexes: NDUFB8 subunit of complex I, SDHB subunit of complex II, core protein 2 UQCRC2 of complex III, MTCO1 subunit 1 of complex IV, and α-subunit of complex V or ATP synthase. For detection of OXPHOS subunits, all western blot steps were followed.

### 2.4. Cell Culture Studies

Cells of the H9c2 rat cardiomyoblast cell line (Merck, Darmstadt, Germany) were maintained in Dulbecco’s modified Eagle’s medium (DMEM; Merck, Darmstadt, Germany) supplemented with 25 mM glucose, 1 mM pyruvate, and 2 mM L-glutamine. Cells were cultured according to the manufacturer′s instructions and were used until passages 15–17. For the experiments, cells were seeded into 6-well plates at 90% confluence, and serum starved for 18 h, and 6 wells were evaluated for each experimental condition. All assays in the present study were done at a temperature of 37 °C, 95% sterile air, and 5% CO_2_ in a saturation humidified incubator. Cells were stimulated with 200 µM of PAwith bovine serum albumin (BSA) as a carrier—conjugated to 10% free fatty acids (FFA)-free BSA for 24 h in the presence or absence of MitoQ (5 nM). In order to evaluate the potential pathways involved in the effects of PA, cells were stimulated with 200 µM of palmitate–BSA for 5–60 min, and total and phosphorylated proteins of Akt, Erk, and NF-κB were evaluated.

### 2.5. Statistical Analysis

Variables are expressed as mean ± SEM. Normality of distributions was verified by means of the Kolmogorov–Smirnov test. Differences among groups were analyzed using 1-way ANOVA followed by Newman-Keuls test. Pearson correlation analysis was used to examine association among different variables according to whether they are normally distributed. A value of *p* < 0.05 was used as the cutoff value for defining statistical significance. Data analysis was performed using the statistical program SPSS version 25.0 (SPSS Inc, Chicago, IL, USA).

## 3. Results

Animals fed an HFD showed an increase in body weight and adiposity index after seven weeks of HFD intake when compared to control rats [12]. Obese rats also showed an increase in relative heart weight, suggesting cardiac hypertrophy, cardiomyocyte area, and cardiac fibrosis, although no changes in cardiac function or blood pressure were observed (Table 1, Figure 1A,B). Treatment with MitoQ reduced the changes observed in obese rats, supporting the role of mitochondrial oxidative stress in these alterations. MitoQ treatment did not affect any of these parameters in control animals (Table 1) [12]. Therefore, and to simplify the results, only rats fed a standard diet and HFD treated with vehicle or MitoQ (HFD + MitoQ) will herein be considered.

The efficiency of MitoQ treatment was evaluated by measurement of both total and mitochondrial cardiac O_2_^−^ levels. HFD-fed rats showed higher O_2_^−^ of both total and mitochondrial cardiac levels as compared with CT animals, as suggested by the higher fluorescence staining observed in the heart of the obese animals in response to DHE or MitoSox, respectively (Figure 1C,D and Appendix A). These higher levels were reduced by the MitoQ treatment (Figure 1C,D and Appendix A), supporting the effectiveness of MitoQ. We then explored whether the increase in oxidative stress can affect myocardial use of glucose. Representative examples of PET images (Figure 1E,F) revealed that the control group show more ^18^-FDG uptake in the hearts as compared with obese animals. The administration of MitoQ was able to ameliorate myocardial ^18^-FDG uptake in HFD-fed rats, although it was unable to normalize it, suggesting the participation of mitochondrial oxidative stress, at least in part, in the cardiac metabolic alterations observed in obese rats. Similarly, MitoQ was able to reduce the increase in HOMA index in obese rats [12]. The ^18^-FDG uptake levels were negatively associated with cardiac fibrosis and hypertrophy (Table 2).

An increase in lipid content was observed in the heart of obese rats (Figure 2A,B) that was accompanied by an increase in total mitochondrial triglycerides (TG) content in obese animals (Figure 2C), mainly due to an accumulation of TGs enriched with PA (C16:0), which account for almost 50% of these (Figure 2D). Treatment with MitoQ was able to reduce this rise observed in animals that were fed an HFD (Figure 2A–D). Cardiac TG levels were associated with those of ^18^-FDG uptake, cardiac fibrosis, and cardiac hypertrophy (Table 2). We explored whether high cardiac mitochondrial TG levels could be due to an increase in flow into mitochondria. Figure 3A shows an increase in CPT1A levels in obese rats as compared with controls, which was prevented by MitoQ treatment. A similar pattern was observed with FAT, a fatty acid protein transport into the cell, since it was increased in obese animals and reduced in those treated with MitoQ (Figure 3B). The mitochondrial cardiac TG levels were associated with both CPT1A (r = 0.674; *p* = 0.001) and FAT (r = 0.588; *p* = 0.008) levels. An increase in DGAT1 levels—involved in the last step of TG synthesis—was also observed in HFD rats and reduced in those treated with MitoQ (Figure 3C). However, no significant differences were observed among any group in the levels of ATL involved in TG degradation (Figure 3D).

Cardiac levels of FFAs were elevated in HFD animals as compared with CT (Figure 4A) mainly due to an accumulation of those enriched with saturated FFA (C16:0, C18:0) or ω-6 FFAs (Appendix A). No significant changes were observed in the levels of those FFAs enriched with ω-3 among any group. Consequently, the ratio ω-6/ω-3 was increased in obese animals (data not shown). MitoQ treatment was able to reduce these increases supporting the participation of mitochondrial oxidative stress in these changes (Figure 4A and Appendix A). Total FFA levels were associated with the levels of those of either cardiac fibrosis or hypertrophy (Table 2). Cardiolipins (CL) levels were reduced in obese animals and normalized in MitoQ-treated rats, supporting the role of mitochondrial oxidative stress in the upregulation of CL (Figure 4B). These levels were negatively associated with those of fibrosis (Table 2).

Total ceramide levels were upregulated in obese rats and were reduced in HFD-treated rats (Figure 4C). However, the very-long-chain C22:0-ceramide dramatically decreased in obese rats and normalized with MitoQ treatment (Appendix A). Total mitochondrial ceramide levels were negatively associated with those of ^18^-FDG uptake and positive with the levels of cardiac fibrosis or cardiac hypertrophy (Table 2).

Considering that there is an accumulation of lipids in the mitochondria of obese animals, we explored the consequences on proteins involved in different mitochondrial functions. To this end, we evaluated the levels of two proteins involved in the process of fusion and fission. As shown in Figure 5A, the protein levels of MFN1 are higher in obese animals as compared with controls. MitoQ treatment was able to reduce these levels. MFN1 levels were negatively associated with those of ^18^-FDG uptake and positive with cardiac hypertrophy (Table 2). Neither obesity nor MitoQ were able to affect the levels of DRP1 (Figure 5B).

Obesity was able to increase Cyclo F levels (Figure 5C), suggesting the opening of the mitochondrial permeability transition pores (mPTP), which was normalized by MitoQ treatment, supporting the role of oxidative stress in this process. Also, an increase of PGC1α levels were reduced by MitoQ treatment (Figure 5D). A reduction in the levels of FH was observed in obese rats, which was normalized by MitoQ treatment, indicating the participation of mitochondrial ROS in this alteration (Figure 5E). These levels were negatively correlated with those of cardiac fibrosis and hypertrophy but positive with those of ^18^-FDG uptake (Table 2). In addition, FH levels were negatively correlated with those of TG and ceramides (*r* = −0.451; *p* = 0.046, *r* = −0.485; *p* = 0.03; respectively). No differences were observed between HFD and CT animals in the levels of other enzymes involved in citric acid cycle ACS, and MitoQ was unable to modify them (Appendix A). Neither obesity nor MitoQ were able to affect the levels of cyto C (Figure 5F). We also evaluated the protein levels of the components of mitochondrial respiratory chain complexes. As shown in Figure 6A–F, obesity exerts a different impact on different complexes, since obese animals show a decrease in complex I and II but not changes in the levels of complex III, IV, and V, the ATP synthase. MitoQ treatment was able to prevent these changes (Figure 6A,B).

Taking the observed accumulation of PA in the mitochondria into consideration, we decided to assess whether PA could directly increase oxidative stress or affect mitochondrial proteins involved in essential functions by using cultured rat cardiomyoblasts. PA induced an increase in mitochondrial oxidative stress, as suggested by the increase in the levels of 4HNE, a major end-product of lipid peroxidation (Figure 7A). Similarly, PA induced an increase in CPT1A levels, although it reduced those of FAT as compared with control conditions (Figure 7B,C, respectively). The presence of MitoQ was only able to prevent the change in 4HNE and CPT1A levels (Figure 7A–C). An increase in levels of both Cyclo F and Cyto C were observed in PA-stimulated cells (Figure 7D,E). MitoQ treatment prevented these changes. No significant changes were observed in response to either PA or MitoQ in the protein levels of FH, PCG1-α, and MFN1 (Figure 8A–C). Neither PA nor the presence of MitoQ in the medium was able to affect the expression of the different mitochondrial complexes in H9c2 cells (Appendix A).

In order to explore the potential pathways involved in the direct actions induced by PA on mitochondrial proteins in H9c2 cells, we explored its effects on Akt, Erk, and NF-κB phosphorylation*s* (Figure 8D–G). PA was able to stimulate the phosphorylation of Erk1/2 in cardiac myoblasts, reaching the maximum level at 10 min, and were elevated until 30 min (Figure 8D). Similarly, PA was able to phosphorylate NF-κB, reaching the maximum level at 60 min (Figure 8E). However, it was unable to affect the Akt pathway (Figure 8F).

## 4. Discussion

Cardiolipotoxicity, as well as mitochondrial oxidative stress, are well-accepted mechanisms underlying cardiac damage in the context of obesity. However, the potential interactions between these mechanisms are a matter of discussion. The results presented here reveal that the increase in total lipids at the cardiac mitochondrial level observed in obese rats was accompanied by an exacerbation of oxidative stress, a reduced myocardial glucose uptake, cardiac fibrosis, myocardial hypertrophy and alterations in the mitochondrial protein in the heart. We postulate that mitochondrial oxidative stress could participate in cardiac lipotoxicity associated with obesity. Overall, we found that the administration of the mitochondrial antioxidant MitoQ protects against some of the consequences of diet-induced obesity, thereby attenuating the disturbances in mitochondrial lipid remodelling, improving cardiac metabolism, reducing cardiac fibrosis and hypertrophy, and improving the alterations in mitochondrial proteins observed in obese animals.

HFD-fed animals exhibited a decline in cardiac glucose uptake, suggesting insulin resistance, a common feature in obese patients [12,18]. This is considered to be an early adaptation to diet-induced obesity [19] and to trigger an imbalance in cell substrate availability. This metabolic maladaptation facilitates progressive lipid accumulation. Along these lines, our data show an increase in intracellular lipid accumulation in the heart. This was accompanied by an increase in mitochondrial TG, especially those enriched with PA in normotensive obese animals, which supports cardiac lipotoxicity, a well-established feature in the context of obesity or diabetes [1,2,20]. This excess in TG seems to have functional consequences because it was associated with cardiac fibrosis and hypertrophy. An increase in their synthesis could be underlying these high TG levels in obese rats, since the increase in DGAT1 protein levels, the rate-limiting enzyme required for the final step of TG synthesis [21], observed in obese animals was normalized in those treated with MitoQ that show normal mitochondrial TG levels. Similar results have been reported in mice [22,23]. By contrast, a reduction in TG degradation seems not to play a major role because no differences were observed in ATL, an enzyme involved in the first step in the intracellular degradation of TG, among any group. This increase in synthesis could be facilitated by a higher FA uptake at both cellular and mitochondrial levels, since a rise in protein levels of FAT and CPT1A, respectively, was observed in obese animals. In fact, a correlation was found between mitochondrial TG levels and those of these proteins. Similar increases have been found and were associated with high cardiac lipid content, not only in the context of obesity, but also in that of diabetes and myocardial ischemia [5,16,24].

An increase in FFAs was also observed in obese rats, confirming previous observations, not only in obesity but also in diabetes [25,26]. This accumulation could involve a higher FA uptake at both cellular and mitochondrial levels, as mentioned above due to increased lipid supply in obesity. An increase in lipogenesis or decreased FA oxidation in the cardiomyocytes could not be eliminated. These high levels are mainly due to saturated and ω-6 FFA with a poorer cardiovascular profile. In fact, not only total levels, but especially the ω-6 FFA, and to a lesser degree, saturated ones, were associated with cardiac fibrosis and cardiac hypertrophy. Similarly, high FFA levels were accompanied by cardiac hypertrophy and fibrosis, not only in obesity, but in other pathological situations [27,28]. In agreement with that, different studies have shown that the quality of FA determines shifts to cardiac damage or cardioprotection [29,30].

Ceramides are not only essential structural components of cell membranes, but also function as second messenger molecules in cell signaling, and have been implicated in different pathological settings [31,32,33]. The present data confirm previous observations, which support a negative effect of ceramides in insulin signaling in different tissues [16,33,34]. In addition, our data supports a role for ceramides in myocardial fibrosis, as well as hypertrophy.

Similarly, there have been reported high cardiac levels of both ceramide and collagen in a model of myocardial infarction in mice [35,36], as well as high ceramide cardiac levels in patients with a failing heart of a different origin [35]. This increase seems to be mainly due to an increase in novo formation [35]. The potential role of ceramides in the development of fibrosis could involve the activation of the transforming growth factor-β signaling pathway [36], which participates in the development of cardiac fibrosis in diet-induced obesity [37]. In addition, high cardiac ceramide levels were observed in cardiac hypertrophied hearts, not only in obesity, but also in other pathological conditions [38,39].

In general, this supports a relevant role of ceramide in the cardiac damage associated with obesity. In this line, it has been shown that the level of circulating ceramides is a possible biomarker for identifying patients at risk for major adverse cardiac events [40]. In fact, an association between plasma ceramides and risk of cardiovascular disease has been reported [41]. Although mitochondrial ceramide levels were elevated in obese rats overall, the levels of C22:0-ceramide—with protective effect against hypoxia [42]—were selectively down-regulated in obese rats. Therefore, the imbalance between the different ceramide species may generate a more harmful cardiac environment in the context of obesity.

Our data also showed that obesity was accompanied by a reduction in total cardiac CL levels, confirming previous findings not only in the context of obesity, but also in other pathological conditions, including diabetes, heart failure, and myocardial ischemia [5,43,44]. This decrease can occur even before TG accumulation [45] and could be a consequence of a reduction in CL synthesis [46]. These lower CL levels were associated with cardiac fibrosis, supporting functional consequences. CL are essential components of mitochondrial membrane that interacts with membrane proteins involved in mitochondrial functions, including some which regulate bioenergetics, membrane structure, and apoptosis [47,48,49]. In this regard, we found a reduction in complexes I and III protein levels, supporting the notion of an impaired oxidative phosphorylation that further supports an impaired energy balance. Also involved in such impairment is a reduction in FH, an enzyme involved in tricarboxylic acid cycle, observed in the hearts of obese animals.

Obese rats showed higher PCG-1α levels than CT animals, supporting an increase in mitochondrial biogenesis. Similarly, obesity was accompanied by higher protein levels of MFN1, thereby suggesting an augmented fusion that adapts the morphology of the mitochondria to the metabolic needs of the cell. These changes could be an adaptation to energy stress, since maintaining mitochondrial mass, or even an enhancement after reduced energy source, permits mitochondria to maximize the cell energy supply [50].

An exacerbation of oxidative stress, mainly from mitochondrial origin, is linked to mitochondrial lipid remodeling and, in consequence, the cardiac damage associated with obesity, since the administration of MitoQ was able to normalize the cardiolipotoxicity, mitochondrial alterations and the cardiac consequences of obesity. A role of oxidative stress has also been reported in the cardiac damage associated, not only with obesity, but also with aging, heart failure and hypertension [10,51,52]. In this regard, myocardial ischemia-reperfusion was accompanied by changes in cardiac metabolism in a model of non-obese and non-diabetic mice with ectopic lipid deposition, leading to severe myocardial dysfunction. These alterations were associated with cardiac mitochondrial alterations and increased ROS production, suggesting the possible role of mitochondrial oxidative stress in these alterations [53]. Mitochondria are the main source of intracellular ROS production, which occurs through electron leakage as a byproduct of ATP generation by oxidative phosphorylation. In addition, the reduced levels of FH can also participate in the increases of fumarate concentration and promote oxidative stress in cardiac cells [54]. The increased oxidative stress can also stimulate the appearance of permeability transition pore (mPTP), an opening that contributes to mitochondrial dysfunction, since it has been shown that the upregulation of cycloF occurs in a ROS-dependent manner [55]. The cycloF induces the opening of mPTP, leading to osmotic imbalance and consequent mitochondrial swelling, which triggers cell death. In addition, further cardiac actions have been proposed for cycloF, including cardiac hypertrophy and fibrosis [56,57].

Our data show that mitochondrial oxidative stress is involved in the consequences of cardiolipotoxicity in the context of obesity by facilitating the lipid remodeling and, therefore, the cardiac consequences observed in obese rats. Moreover, the data show that PA, the main FA accumulated in the heart of obese rats, is able to stimulate oxidative stress in H9C2 cells and facilitates some of the alterations observed in the heart of obese rats, thereby closing a vicious circle. The presence of PA increased the protein levels of CPT1A, supporting the notion of a higher FA uptake at mitochondrial levels, although a reduction in those of FAT could be a compensatory mechanism to excess exposure to FFAs. PA also induced the appearance of mPTP, as suggested by the increase in cyclopF protein levels. Such an increase was accompanied by a rise in Cyto C levels, a fact previously reported in H9c2 cells in response to PA [58]. However, we did not evaluate where these high levels were located, although they could be expected to be cytoplasm since it can escape through mPTP, thereby reducing mitochondrial levels [58]. MitoQ ameliorated the increase in CPT1A, CycloF, and Cyto C induced by PA in these cells, in agreement with our in vivo data. It has consistently been reported that PA interferes with several signaling pathways, including the Erk and NF-κB pathways [58,59]. Therefore, these data suggest interactions between lipotoxicity and mitochondrial oxidative stress, which regulate downstream events responsible for cardiac alterations in the context of obesity.

The direct exposition of H9C2 cells to PA was unable to affect other mitochondrial proteins involved in relevant functions that were affected in obese rats, including MFN1, PGC1α, or FH. A similar situation was found with the mitochondrial complexes, supporting that factor or factors activated by the complex environment that occurs in the context of obesity could be necessary to affect these mitochondrial proteins.

In summary, to understand the interactions among the mechanisms that underlie the cardiac damage in the context of obesity is crucial. Our present data show the cardiac alterations observed in obese rats were accompanied by an important remodeling in the mitochondrial lipids that can be facilitated by the excess of mitochondrial oxidative stress. In addition, PA was able to stimulate oxidative stress in cardiac myoblasts and altered some of the proteins involved in essential functions of the mitochondria observed in the obese animals. Therefore, the data suggest a crosstalk between lipotoxicity and mitochondrial oxidative stress, drawing a more complex scenario of the role of mitochondria in cardiac damage in the context of obesity and suggest new approaches in the management of obesity-related heart damage.

## Figures and Tables

**Figure 1 cells-09-00451-f001:**
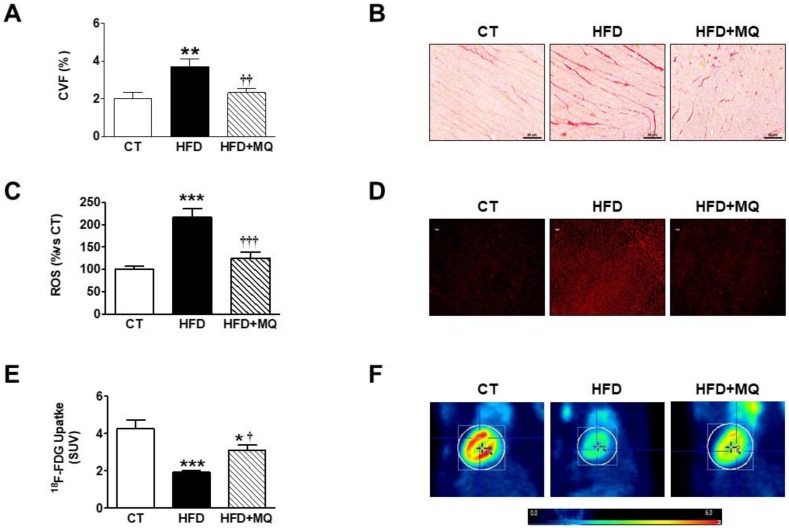
Effects of the mitochondrial antioxidant MitoQ (MQ) on cardiac interstitial fibrosis, superoxide anion levels and cardiac ^18^-FDG uptake in obese rats. Hearts from rats fed a standard diet (CT) or a high fat diet (HFD) treated with vehicle or with MQ (200 µM) were analyzed. (**A**) Quantification of collagen volume fraction (CVF) and (**B**) representative microphotographs of sections stained with picrosirius red. (**C**) Quantification of total cardiac levels of superoxide anion and (**D**) representative microphotographs of cardiac sections labeled with DHE. (**E**) Quantification of myocardial ^18^-FDG uptake and (**F**) Representative photographs of ^18^F-Fluorodeoxyglucose (FDG) PET/CT scans in trans-axial views scaled to standardized uptake value (SUV). Magnification (×40).Bar graphs represent the mean ± SEM. of 6–8 animals. * *p* < 0.05; ** *p* < 0.01; *** *p* < 0.001 vs. control group. ^†^
*p* < 0.05; ^††^
*p* < 0.01; ^†††^
*p* < 0.001 vs. HFD group.

**Figure 2 cells-09-00451-f002:**
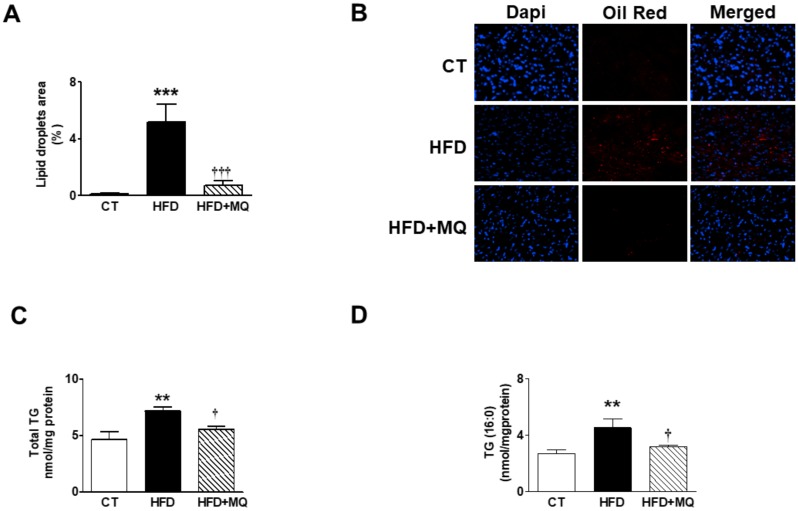
Impact of the mitochondrial antioxidant MitoQ (MQ) on lipid droplet content and mitochondrial triglycerides in the hearts of obese rats. Hearts from rats fed a standard diet (CT) or a high fat diet (HFD) treated with vehicle or with MQ (200 µM) were analyzed. (**A**) Quantification of intracellular lipid accumulation and (**B**) representative microphotographs of sections stained with DAPI and Oil red O to visualize and quantify nuclei and intracellular lipid droplets, respectively. (**C**) Levels of total mitochondrial triglycerides (TGs) and (**D**) mitochondrial TGs enriched with palmitic acid (PA; C16:0) are presented. Magnification (×40). Bar graphs represent the mean ± SEM. of 6–8 animals. ** *p* < 0.01; *** *p* < 0.001 vs. control group. ^†^
*p* < 0.05; ^†††^
*p* < 0.001 vs. HFD group.

**Figure 3 cells-09-00451-f003:**
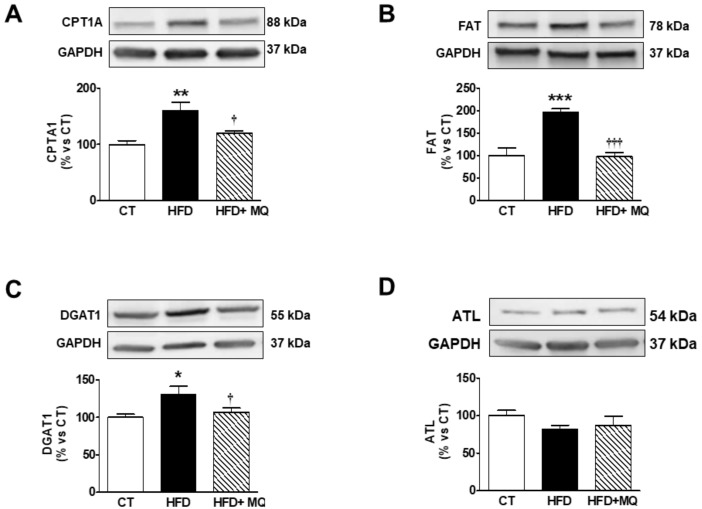
Effects of the mitochondrial antioxidant MitoQ (MQ) on protein levels in the hearts of obese rats. Hearts from rats fed a standard diet (CT) or a high fat diet (HFD) treated with vehicle or with MQ (200 µM) were analyzed. Protein expression of (**A**) carnitine palmitoyltransferase 1 (CPT1A), (**B**) fatty acid translocase (FAT), (**C**) diacylglycerol transferase 1 (DAGT1) and (**D**) adipose triglyceride lipase (ATL) are presented. Bar graphs represent the mean ± SEM of 6–8 animals normalized to glyceraldehyde-3-phosphate dehydrogenase (GAPDH). * *p* < 0.05; ** *p* < 0.01; *** *p* < 0.001 vs. control group. ^†^
*p* < 0.05; ^†††^
*p* < 0.001 vs. HFD group.

**Figure 4 cells-09-00451-f004:**
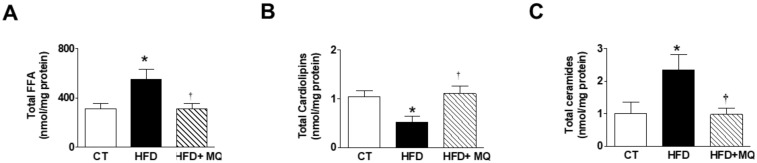
Effects the mitochondrial antioxidant MitoQ (MQ) on mitochondrial lipid species in the hearts from obese rats. Hearts from rats fed a standard diet (CT) or a high fat diet (HFD) treated with vehicle or with MQ (200 µM) were analyzed. Levels of (**A**) mitochondrial free fatty acids (FFA), (**B**) cardiolipins, and (**C**) mitochondrial ceramides are presented. Bar graphs represent the mean ± SEM of 6–8 animals * *p* < 0.05 vs. control group. ^†^
*p* < 0.05 vs. HFD group.

**Figure 5 cells-09-00451-f005:**
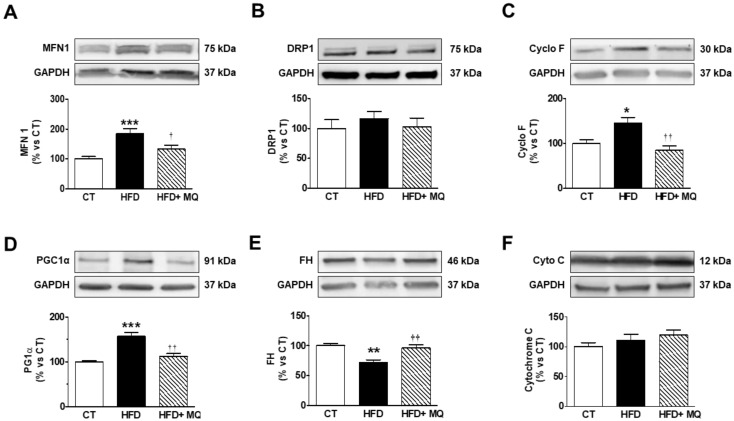
Impact of the mitochondrial antioxidant MitoQ (MQ) on mitochondrial protein levels in the hearts of obese rats. Hearts from rats fed a standard diet (CT) or a high fat diet (HFD) treated with vehicle or with MQ (200 µM) were analyzed. Protein expression of (**A**) mitofusin 1 (MFN1), (**B**) Dynamin-1-like protein (DRP1), (**C**) cyclophilin F (Cyclo F), (**D**) peroxisome proliferator-activated receptor gamma coactivator 1-alpha (PGC-1α), (**E**) fumarate hydratase (FH) and (**F**) cytochrome C are presented. Bar graphs represent the mean ± SEM. of 6–8 animals normalized to glyceraldehyde-3-phosphate dehydrogenase (GAPDH). * *p* < 0.05; ** *p* < 0.01; *** *p* < 0.001 vs. control group. ^†^
*p* < 0.05; ^††^
*p* < 0.01 vs. HFD group.

**Figure 6 cells-09-00451-f006:**
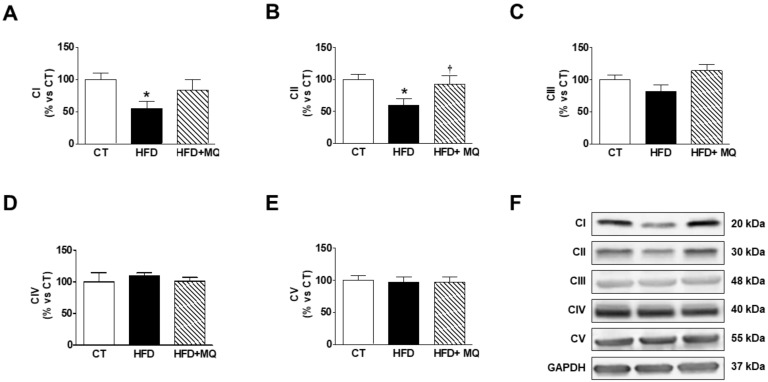
Effects of the mitochondrial antioxidant MitoQ (MQ) on mitochondrial complexes protein levels in the hearts of obese rats. Hearts from rats fed a standard diet (CT) or a high fat diet (HFD) treated with vehicle or with MQ (200 µM) were analyzed. Protein expression of mitochondrial complexes (**A**) I (subunit NDUFB8), (**B**) II (SDHB subunit of complex II), (**C**) III (core protein 2 UQCRC2), (**D**) IV (MTCO1 subunit 1), and (**E**) V (alpha subunit) are presented. (**F**) Representative blots for mitochondrial complexes protein expressions. Bar graphs represent the mean ± SEM. of 6–8 animals normalized to glyceraldehyde-3-phosphate dehydrogenase (GAPDH). * *p* < 0.05 vs. control group. ^†^
*p* < 0.05 vs. HFD group.

**Figure 7 cells-09-00451-f007:**
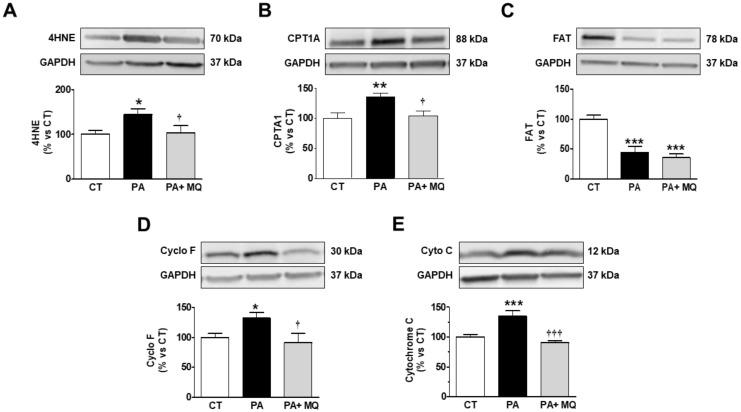
Effects of antioxidant MitoQ (MQ) on protein levels in palmitic acid (PA)-treated H9c2 cells. Protein levels of (**A**) 4-hydroxynonenal (4HNE), (**B**) carnitine palmitoyltransferase 1 (CPT1A), (**C**) fatty acid translocase (FAT), (**D**) cyclophilin F (Cyclo F), and (**E**) cytochrome C in cardiac myoblasts treated for 24 h with PA (200 µM) in the presence of absence of the mitochondrial antioxidant MitoQ (MQ; 5 nM). Bar graphs represent the mean ± SEM of four to six assays normalized to glyceraldehyde-3-phosphate dehydrogenase (GAPDH). * *p* < 0.05; ** *p* <0.01; *** *p* <0.001 vs. control conditions. ^†^
*p* < 0.05; ^†††^
*p* < 0.001 vs. PA-treated cells.

**Figure 8 cells-09-00451-f008:**
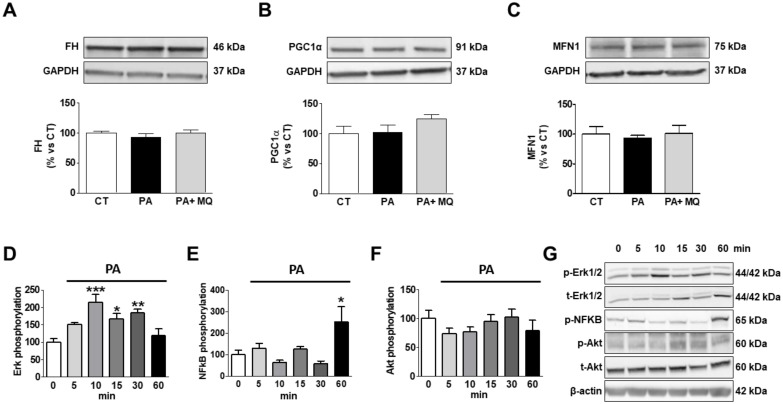
Effects of antioxidant MitoQ (MQ) on protein levels in palmitic acid (PA)-treated H9c2 cells. Protein levels of (**A**) fumarate hydratase (FH), (**B**) peroxisome proliferator-activated receptor gamma coactivator 1-alpha (PGC-1α), (**C**) mitofusin 1 (MFN1) in cardiac myoblasts treated for 24 h with PA (200 µM) in the presence of absence of the mitochondrial antioxidant MitoQ (MQ; 5 nM). (**D**) Phosphorylated protein of extracellular signal-regulated kinases (pErk1/2), (**E**) phosphorylated protein of nuclear factor-κB p65 (pNF-κB), and (**F**) phosphorylated protein kinase B (pAkt) in cardiac myoblasts treated with PA (200 µM) at different times. (**G**) Representative images blots for phosphorylation of intracellular signaling pathways. Bar graphs represent the mean ± SEM of four to six assays normalized to glyceraldehyde-3-phosphate dehydrogenase (**A**–**C**), β-actin, (**E**) or total Erk1/2 (**D**) and total Akt (**F**). * *p* < 0.05; ** *p* < 0.01; *** *p* < 0.001 vs. control conditions.

**Table 1 cells-09-00451-t001:** Effect of the antioxidant mitochondrial MitoQ (200 µM) on relative heart weight, cardiomyocyte cross-sectional area, echocardiographic parameters, and systolic blood pressure in rats fed a normal chow (CT) and rats fed a high fat diet (HFD).

	CT	MitoQ	HFD	HFD-MitoQ
HW/TL (g/cm)	0.20 ± 0.003	0.21 ± 0.006	0.24 ± 0.007 ***	0.22 ± 0.004 *^†^
CSA (µm^2^)	301.2 ± 7.8	308.1 ± 7.7	380.3 ± 11.9 ***	322.7 ± 10.8 ^†††^
IVTs (mm)	0.130 ± 0.008	0.137 ± 0.006	0.143 ± 0.02	0.135 ± 0.008
PWTs (mm)	0.190 ± 0.013	0.140 ± 0.008	0.171 ± 0.012	0.167 ± 0.019
ESD (mm)	0.352 ± 0.034	0.403 ± 0.012	0.293 ± 0.030	0.349 ± 0.032
EDD (mm)	0.653 ± 0.043	0.670 ± 0.022	0.615 ± 0.023	0.613 ± 0.032
EF (%)	77.8 ± 3.7	73.2 ± 2.8	82.3 ± 1.9	77.3 ± 5.3
FS (%)	44.0 ± 3.3	41.4 ± 2.0	49.3 ± 4.7	43.2 ± 4.8
E/A	1.05 ± 0.08	1.20 ± 0.03	1.07 ± 0.05	1.20 ± 0.02
SBP (mmHg)	133.7 ± 2.5	130.6 ± 3.9	138.1 ± 2.1	133.9 ± 2.6

Data values represent mean ± SEM of 8 animals. CSA indicates cardiomyocyte cross-sectional area; EDD, end-diastolic diameter; EF, ejection fraction; ESD, end-systolic diameter; FS, fractional shortening; HW, heart weight; IVT, interventricular septum thickness; PWT, posterior wall thickness; SBP, systolic blood pressure; and TL, tibia length. * *p* < 0.05, *** *p* < 0.001 vs. control group and ^†^
*p* < 0.05; ^†††^
*p* < 0.001 vs. HFD group.

**Table 2 cells-09-00451-t002:** Association between cardiac fibrosis, hypertrophy, and ^18^-FDG uptake, and those of mitochondrial lipid composition and proteins in the hearts of rats fed either a normal chow (CT) or a high fat diet (HFD) treated with vehicle or MitoQ (200 µM).

	Cardiac Fibrosis		Cardiac Hypertrophy		^18^-FDG Uptake	
	*r*	*p*	*r*	*p*	*r*	*p*
Cardiac fibrosis	-	-	0.792	<0.0001	−0.636	0.003
Cardiac hypertrophy	0.792	<0.0001	-	-	−0.661	0.002
Mitochondrial TGs	0.586	0.007	0.641	0.003	−0.578	0.01
Mitochondrial FFA	0.759	<0.0001	0.487	0.048	-	-
Mitochondrial CL	−0.555	0.017	-	-	-	-
Mitochondrial Ceramides	0.547	0.015	0.624	0.006	−0.460	0.041
MFN1	-	-	0.549	0.018	−0.581	0.011
FH	−0.564	0.01	−0.469	0.037	−0.596	0.006

CL: cardiolipins, FFA: free fatty acid, FH: fumarate hydratase, MFN1: mitofusin 1; TGs: triglycerides.

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
