# Peer review of "The Crosstalk between Cardiac Lipotoxicity and Mitochondrial Oxidative Stress in the Cardiac Alterations in Diet-Induced Obesity in Rats"

_cells, 2020, doi:10.3390/cells9020451_

Round 1

Reviewer 1 Report

I congratulate the Authors for an excellent work on the role of lipotoxicity and mitochondrial oxidative stress in the process of cardiac damage.

Minor remarks:

The manuscript requires some English editing to make it even of higher value for the readers. A few examples: in the Introduction in the sentence “Mitochondria comprise almost a third of cardiomyocyte volume, is the source of ATP and plays a central role …...” mitochondria are used as plural form therefore it should be written “ … are the source of ATP and play a central role…”. Then “Mitochondria is also the main source …” should be rather “Mitochondria are also the main source…”. In the text the Authors use the term  “western blot” writing it with capital letter (“Western blot”) whereas it should be written with a small letter – the technique is not named after its inventor (as it is for example in case of Southern blot named after Edwin Southern).

Author Response

I congratulate the Authors for an excellent work on the role of lipotoxicity and mitochondrial oxidative stress in the process of cardiac damage.

Thank you very much for your comments. We have made every effort to take them into consideration when correcting the text (see below and the responses to the reviewer).

Minor remarks:

The manuscript requires some English editing to make it even of higher value for the readers. A few examples: in the Introduction in the sentence “Mitochondria comprise almost a third of cardiomyocyte volume, is the source of ATP and plays a central role …...” mitochondria are used as plural form therefore it should be written “ … are the source of ATP and play a central role…”. Then “Mitochondria is also the main source …” should be rather “Mitochondria are also the main source…”. In the text the Authors use the term  “western blot” writing it with capital letter (“Western blot”) whereas it should be written with a small letter – the technique is not named after its inventor (as it is for example in case of Southern blot named after Edwin Southern).

Reply and changes:

We thank the reviewer for this appreciation. We have corrected the errors pointed out by the referees.

“Mitochondria comprise almost a third of cardiomyocyte volume, is the source of ATP and plays a central role…”

Page 2, lines 54-55.

 “Mitochondria is also the main source …” should be rather “Mitochondria are also the main source…”.

Page 2, lines 56-57.

“western blot” writing it with capital letter (“Western blot”) whereas it should be written with a small letter “

Page 5, line 216.

In addition, a complete English editing of the manuscript has been performed by a native English speaker.   

Reviewer 2 Report

"THE CROSSTALK BETWEEN CARDIAC LIPOTOXICITY AND MITOCHONDRIAL OXIDATIVE STRESS IN THE CARDIAC ALTERATIONS IN DIET-INDUCED OBESITY IN RATS" by Jiménez-González and colleagues is a very interesting, well executed study in adult rats on the effects of excess lipid accumulation on mitochondrial stress and cardiac fibrosis. It only requires minor spell/grammar checks. I am curious about two issues, one of which the authors can easily address-the second one maybe not so much.   1). The sex distribution of the rats in the three groups for which data is presented is not given and should be added to the methods section. Were these all male, all female, or 1:1 ratio of both sexes in the groups? Was there any sex differences noted IF both sexes were equally represented. 2). The authors give pretty comprehensive echocardiographic data on systolic parameters like ejection fraction. Were any doppler imaging/strain/tissue doppler imaging performed to assess diastolic function? Even though EF was not different between the groups, there was more hypertrophy and cardiac fibrosis in the obese rats which is likely to translate into diastolic dysfunction, st least if given enough time (7 weeks may or may not be enough time).

Author Response

"THE CROSSTALK BETWEEN CARDIAC LIPOTOXICITY AND MITOCHONDRIAL OXIDATIVE STRESS IN THE CARDIAC ALTERATIONS IN DIET-INDUCED OBESITY IN RATS" by Jiménez-González and colleagues is a very interesting, well executed study in adult rats on the effects of excess lipid accumulation on mitochondrial stress and cardiac fibrosis. It only requires minor spell/grammar checks. I am curious about two issues, one of which the authors can easily address-the second one maybe not so much.  

Thank you very much for your comments. We have made every effort to take them into consideration when correcting the text (see below and the responses to the reviewer).

The sex distribution of the rats in the three groups for which data is presented is not given and should be added to the methods section. Were these all male, all female, or 1:1 ratio of both sexes in the groups? Was there any sex differences noted IF both sexes were equally represented.

Reply and changes:

Although we understand that comparison between both sexes could give a more comprehensive view of how cardiac lipotoxicity and mitochondrial oxidative stress interact in the cardiac alterations in diet-induced obesity, only male rats were used in the study. The decision was based on the fact that estrogens exert a modulation of the cardiovascular system. It is thus necessary to control the estrous cycle in the female rats before any measurement for comparing animals at the same moment of the cycle. This situation made it more complex to perform the different parameters, especially those that need to be made within a very specific time frame (i.e. cardiac function and PET studies). The revised manuscript has been modified in order to clarify this point in the abstract as well as in material and methods.

 The authors give pretty comprehensive echocardiographic data on systolic parameters like ejection fraction. Were any doppler imaging/strain/tissue doppler imaging performed to assess diastolic function? Even though EF was not different between the groups, there was more hypertrophy and cardiac fibrosis in the obese rats which is likely to translate into diastolic dysfunction, at least if given enough time (7 weeks may or may not be enough time). 

Reply and changes:

As indicated in the Material and Methods, diastolic function was only assessed by early and late transmitral peak diastolic flow velocity (E and A waves) through pulsed-wave Doppler recording. Please find below a representative figure of Doppler imaging acquisitions, including recording and measurement of E and A mitral inflow waves (Figure 1).

The ratio between E-waves and A-wave (E/A) was calculated and shown in Table 1.  As pointed out by the reviewer, no differences were observed in diastolic function between obese and control animals even in the presence of cardiac hypertrophy and fibrosis in obese animals. These alterations in obese animals show that structural changes and cardiac remodeling are prior to the onset of cardiac dysfunction and if persist could promote cardiac dysfunction. The lack of differences between both groups in cardiac function could rely on the time of evolution of the obesity. This could be inferred due to the fact that we found differences between obese and control animals in E/A ratio in another study in which the high fat diet was administered to animals for 10 weeks (Figure 2) confirming that the high fat diet maintained over time promotes diastolic cardiac dysfunction (data not shown in the manuscript).
